# MARCKS Inhibition Alters Bovine Neutrophil Responses to *Salmonella* Typhimurium

**DOI:** 10.3390/biomedicines12020442

**Published:** 2024-02-16

**Authors:** Haleigh E. Conley, Chalise F. Brown, Trina L. Westerman, Johanna R. Elfenbein, M. Katie Sheats

**Affiliations:** 1Department of Clinical Sciences, College of Veterinary Medicine, North Carolina State University, Raleigh, NC 27607, USA; 2Comparative Medicine Institute, North Carolina State University, Raleigh, NC 27607, USA; 3Department of Pathobiological Sciences, School of Veterinary Medicine, University of Wisconsin-Madison, Madison, WI 53706, USA

**Keywords:** neutrophils, *Salmonella*, MARCKS, respiratory burst, migration, phagocytosis

## Abstract

Neutrophils are innate immune cells that respond quickly to sites of bacterial infection and play an essential role in host defense. Interestingly, some bacterial pathogens benefit from exuberant neutrophil inflammation. *Salmonella* is one such pathogen that can utilize the toxic mediators released by neutrophils to colonize the intestine and cause enterocolitis. Because neutrophils can aid gut colonization during *Salmonella* infection, neutrophils represent a potential host-directed therapeutic target. Myristoylated alanine-rich C-kinase substrate (MARCKS) is an actin-binding protein that plays an essential role in many neutrophil effector responses. We hypothesized that inhibition of MARCKS protein would alter bovine neutrophil responses to *Salmonella* Typhimurium (STm) ex vivo. We used a MARCKS inhibitor peptide to investigate the role of MARCKS in neutrophil responses to STm. This study demonstrates that MARCKS inhibition attenuated STm-induced neutrophil adhesion and chemotaxis. Interestingly, MARCKS inhibition also enhanced neutrophil phagocytosis and respiratory burst in response to STm. This is the first report describing the role of MARCKS protein in neutrophil antibacterial responses.

## 1. Introduction

Non-typhoidal salmonellae (NTS) are the leading cause of human bacterial foodborne gastroenteritis. Annually, NTS cause more than 150 million illnesses worldwide and the highest number of food-borne-disease-related deaths in the United States [1,2,3,4]. Infection can result from contaminated food or water or direct contact with individuals or animals shedding the pathogen [5]. Despite efforts to decrease the number of infections, the incidence of NTS illnesses has not changed over the past 20 years [6]. In addition to increasing infections, The World Health Organization reports that the incidence of *Salmonella* antimicrobial resistance (AMR) is increasing. The high rate of disease, increasing antimicrobial resistance, and high degree of morbidity and mortality associated with multiple-drug resistant (MDR) NTS infection drive the need to develop new interventions to combat this important pathogen [7,8].

Neutrophils are innate immune cells that play an important role in the pathophysiology of NTS gastroenteritis. Neutrophil responses to bacterial infection in the host include endothelial cell adhesion, chemotaxis and transmigration, bacterial phagocytosis, and production of reactive oxygen species (ROS) [9,10,11]. For most bacterial infections, neutrophil responses facilitate the elimination of the invading pathogen; however, the opposite is true in the case of enteric salmonellosis. While the neutrophilic response limits NTS systemic dissemination, it paradoxically facilitates NTS replication in the gut. *Salmonella* are unique because they can benefit from the neutrophilic inflammatory response locally in the gut and have even been shown to target neutrophils during intestinal infection [12]. NTS withstand neutrophil toxic mediators and utilize these mediators for growth, allowing them to outcompete resident microbiota [13,14,15,16,17]. NTS survival perpetuates neutrophil cytokine and ROS production, leading to neutrophilic enterocolitis [18,19] and a cycle of intestinal inflammation that causes significant injury to host tissue [13,15,20]. These data suggest that a novel strategy to decrease NTS success in the intestine would be to dampen the host neutrophil response.

Myristoylated alanine-rich C-kinase substrate (MARCKS) is a ubiquitously expressed actin-binding protein that plays an essential role in neutrophil functions including adhesion, migration, and respiratory burst [21,22,23,24]. MARCKS alters these processes through a reversible binding of phosphatidylinositol 4,5-bisphosphate (PIP2) and actin.

When dephosphorylated, MARCKS is anchored to the plasma membrane through hydrophobic insertion of the N-terminal myristoyl-moiety into the plasma membrane and through electrostatic interaction of the polybasic effector domain (ED) with the negatively charged phospholipid PIP2 [22,25]. Dephosphorylated MARCKS sequesters PIP2 within the membrane and facilitates actin polymerization and crosslinking [22,26]. On the other hand, phosphorylated MARCKS is displaced to the cytosol, which increases PIP2 availability at the plasma membrane and decreases MARCKS–actin binding [26,27].

In addition to cellular actin regulation, MARCKS also has a clear role in inflammatory responses. MARCKS inhibition with N-terminus or phosphorylation-blocking peptides inhibits neutrophil adhesion, migration, ROS production, and pro-inflammatory cytokine production ex vivo [28,29,30,31]. MARCKS-targeting peptides have also been shown to inhibit neutrophils in animal models and human patients in vivo [32,33,34,35,36,37]. Despite significant interest in MARCKS as a novel anti-inflammatory target, little is known regarding the role of MARCKS in neutrophil responses to bacteria. To address this gap in knowledge, and to obtain proof of concept data on MARCKS as a potential therapeutic target for *Salmonella* enterocolitis, we investigated the impact of MARCKS-inhibiting peptides on bovine neutrophil inflammatory responses to STm ex vivo.

## 2. Materials and Methods

### 2.1. Reagents

Roswell Park Memorial Institute (RPMI) Medium 1640 and heat-inactivated (HI) fetal bovine serum (FBS), Dead Cell Apoptosis kit, and UltraPure 0.5 M EDTA were from Thermo Fisher Scientific (Waltham, MA, USA). Hank’s Balanced Salt Solution (HBSS), Calcein AM, BD Difco LB Broth Miller, BD Difco LB Agar Miller were from Fisher Scientific (Hampton, NH, USA). Recombinant Bovine granulocyte-macrophage colony-stimulating factor (GM-CSF) was from Kingfisher Biotech (Saint Paul, MN, USA). Ficoll (1.077 g/mL) was from Cytiva (previously GE Healthcare, Uppsala, Sweden). Bovine serum (B9433), Dihydrorhodamine 123 (DHR-123), Diphenyleneiodonium chloride (DPI), and Interleukin-8 (IL-8) were from Millipore Sigma (Burlington, MA, USA). 

### 2.2. Peptide Treatment

The MyristoylAted N-terminal Sequence (MANS) and Random Nucleotide Sequence (RNS) peptides were synthesized by Genemed Synthesis, Inc. (San Antonio, TX, USA). The sequence of MANS is identical to the first 24 amino acids of the human MARCKS protein: myristic acid-GAQFSKTAAKGEAAAERPGEAAVA. The RNS peptide is a randomly scrambled control: myristic acid-GTPAPAAEGAGAEVKRASAEAKQAF. The Effector Domain (ED) and Control Peptide (CP) peptides were synthesized by Sigma Aldrich. The sequence of ED with C-terminal internalization TAT sequence: KKKKKRFSFKKSFKLSGFSFKKNKKGRKKRRQRRRPQ. CP is a scrambled version of full-length MARCKS with C-terminal TAT sequence: CEIEEHAWNTVEMFSSFPGTQLYNDAGRKKRRQRRRPQ. Peptide working solutions were resuspended in sterile PBS. Where indicated, pretreatment of cell suspensions with indicated peptide concentrations occurred at 37 °C for 30 min. 

### 2.3. Bovine Neutrophil Isolation

Institutional Animal Care and Use Committees of North Carolina State University (NCSU IACUC protocol #20-116) and the University of Wisconsin-Madison (UW-Madison IACUC protocol #V006249) approved all animal studies. University-owned cows at NCSU and UW-Madison (STm killing only) were blood donors for this study. Cow health was established by veterinarian exam, milk production, and no evidence of inflammatory or infectious disease. All animal experiments were performed in accordance with the PHS “Guide for the Care and Use of Laboratory Animals” in AAALAC-approved facilities.

Jugular or tail vein blood was collected with 1.5 mg/mL EDTA-loaded syringes and 25 mL or less placed in a 50 mL conical. An equal amount of PBS was added to each conical and carefully mixed by inversion. An amount of 30 mL of diluted blood was then layered on top of 15 mL Ficoll (1.077 g/mL). The gradient was centrifuged at 1100× *g* for 30 min at 10 °C with the brake off. The supernatant was removed via aspiration. Hypotonic lysis was carried out by adding 20 mL cold 0.2% NaCl to the RBC/granulocyte pellet. The tube was mixed by gentle inversion and allowed to lyse for 30–40 s. Then, 20 mL cold 1.6% NaCl was added, and tubes were inverted to mix. The tubes were then centrifuged at 100× *g* for 8 min at 4 °C with the brake on. The supernatants were gently removed via aspiration. The lysis and centrifugation steps were repeated for a second time. Neutrophils were resuspended in HBSS and counted via hemocytometer. Trypan blue expression was used to determine percent viability (on average >95%). Cytology slides were prepared and stained with Wright Giemsa to determine the percentage of neutrophils from donor cows at NCSU. Cell counts for experiments were determined by hemocytometer cell counts and neutrophil percentage based on cytology.

### 2.4. Pooled Whole Serum Preparation

Whole blood was collected via the tail vein from 10 lactating dairy cows in the NCSU dairy herd and placed in anti-coagulant-free serum tubes. Tubes were allowed to settle for 1 h prior to centrifugation at 2000× *g* for 10 min at 4 °C. Serum was isolated and pooled prior to the preparation of frozen aliquots. Pooled serum was used as indicated for experiments. 

### 2.5. Bacterial Strains and Growth Conditions

A spontaneous nalidixic acid-resistant clone of *Salmonella enterica* serotype Typhimurium (STm) ATCC (Manassas, VA, USA) 14028.s was used for all experiments [38]. Bacteria containing a plasmid encoding constitutively expressed green fluorescent protein (pTurboGFP-B; Evrogen) were used for phagocytosis [39]. Bacteria were grown on Luria Bertani (LB) agar or in LB broth at 37 °C with agitation (225 rpm). Media was supplemented with the following antibiotics as appropriate: nalidixic acid (50 mg/L) and carbenicillin (100 mg/L). For all neutrophil stimulations, bacteria were grown to a late exponential phase by diluting overnight cultures 1:100 into LB broth and incubating at 37 °C with agitation for 3.5 h [40]. Bacterial cultures were washed in phosphate-buffered saline (PBS), and cell density was estimated by optical density (600 nm; OD_600_). Bacteria were diluted in PBS or indicated buffer and maintained on ice until use. Where indicated, bacteria were opsonized as detailed in individual experiments. Diluted bacteria were plated to establish the number of viable colony-forming units (CFU). 

### 2.6. Propidium Iodide Staining

Isolated primary neutrophils were rested for 1 h at room temperature in HBSS. Neutrophils were primed with 10 ng/mL bovine GM-CSF in HBSS at 1 × 10^7^ cells/mL for 30 min at room temperature protected from light. Neutrophils were centrifuged at 100× *g* for 8 min then resuspended in RPMI supplemented with 1 mM CaCl_2_, 1 mM MgCl_2_, and 2% HI FBS at 1.0 × 10^6^ cells/mL. Neutrophils (100 µL) were added to Eppendorf tubes containing the indicated concentrations of MARCKS-targeting peptides or PBS. Neutrophils were incubated for 90 min at 37 °C with 5% CO_2_, immediately placed on ice, and 1 volume of cold PBS was added. Neutrophils were centrifuged at 300× *g* for 5 min at 4 °C. Samples were stained with 1 µL of 100 µg/mL propidium iodide following the manufacturer’s recommendations (Invitrogen Dead Cell Apoptosis kit, Thermo Fisher) and immediately acquired on a BD LSRII cytometer (BD, Lake Franklin, NJ, USA). The neutrophil population was gated by a forward-scatter versus side-scatter plot to discriminate from cell debris. Singlet cells were included by gating based on side-scatter area versus side-scatter width and forward-scatter area versus forward-scatter width. Data from at least 10,000 events gated on singlet neutrophils were collected. Flow cytometry experiments were performed in the Flow Cytometry and Cell Sorting facility at North Carolina State University—College of Veterinary Medicine.

### 2.7. Neutrophil Adhesion

Immulon2HB clear 96 well plates were coated overnight with 5% HI FBS in HBSS and washed once with PBS prior to experiment. Isolated neutrophils were resuspended to 1 × 10^7^ cells/mL in HBSS, primed with 10 ng/mL GM-CSF, and loaded with 2 µg/mL Calcein AM for 30 min at room temperature protected from light. The neutrophils were then centrifuged at 100× *g* for 8 min, the supernatant removed by aspiration, then resuspended in HBSS supplemented with 1 mM CaCl_2_, 1 mM MgCl_2_, and 2% HI FBS at 1 × 10^6^ cells/mL. Neutrophils were then pretreated in Eppendorf tubes with MARCKS inhibitor peptides as indicated for 30 min at 37 °C with 5% CO_2_. Pretreated neutrophils (100,000) were plated in the wells and allowed to settle for 10 min at room temperature, protected from light. PBS or late exponential-phase STm (MOI 50:1) was added to the neutrophils. The plate was then incubated for 1 h at 37 °C with 5% CO_2_. After the incubation, the fluorescence was measured using a microplate reader at 485 nm excitation and 525 nm emission (Biotek, Winooski, VT, USA). The neutrophils and STm were dumped from the plate, and the wells were washed with PBS to remove non-adherent cells, reading fluorescence at each wash step. The fluorescence values for each wash were divided by initial fluorescence and multiplied by 100 to calculate percent adhesion. The first wash that demonstrated less than 10% adhesion in the non-stimulated neutrophils (plated on 5% FBS coating) was considered the resulting percent adhesion.

### 2.8. Chemotaxis

Isolated primary neutrophils were rested for 1.5 hours at room temperature in HBSS. Neutrophils were primed with 10 ng/mL bovine GM-CSF and loaded with calcein at 2 µg/mL in HBSS at 1 × 10^7^ cells/mL for 30 min at room temperature protected from light. Neutrophils were centrifuged at 100× *g* for 8 min then resuspended in HBSS supplemented with 1 mM MgCl_2_, 1 mM CaCl_2_, and 2% HI FBS (chemotaxis buffer) at 2.2 × 10^6^ cells/mL and pretreated with indicated concentrations of inhibitory peptides or controls for 30 min at 37 °C with 5% CO_2_. For this assay, STm was diluted to achieve an MOI of 50:1 based on OD_600_, centrifuged, then resuspended in chemotaxis buffer and stored at 4 °C until the ChemoTx plate was set up. Chemoattractants (IL-8 and STm) (30 µL) were added in triplicate to the bottom wells of a standard 96-well PCTE ChemoTx plate (5 µm pore size, well capacity 30 µL, 3.2 mm cell site diameter) (NeuroProbe, Gaithersburg, MD, USA). Control (100% migration) wells consisted of 50,000 neutrophils to represent the fluorescence if all neutrophils had migrated. A 5 µm membrane overlaid the chemoattractant-filled bottom wells, and 50,000 neutrophils were added to the membrane. The ChemoTx plate was incubated for 60 min at 37 °C with 5% CO_2_. The non-migrated neutrophils were then removed from the top of the membrane using a cell scraper, and 10 µL 0.5 M EDTA was added to the top of the membrane for 10 min at room temperature protected from light. The excess EDTA was then removed using a cell scraper, and the ChemoTx plate was centrifuged for 5 min at 1000× rpm. The membrane was removed, and the fluorescence of the bottom wells was measured (485 nm excitation, 525 nm emission) using a Biotek microplate reader. Percent migration was calculated by dividing the fluorescence of the experimental bottom wells by the fluorescence of bottom wells containing the known cell number (100% migration group) [24]. 

### 2.9. Phagocytosis—Flow Cytometry

Neutrophils were primed with 10 ng/mL bovine GM-CSF in HBSS at 1 × 10^7^ cells/mL for 30 min at room temperature protected from light. Neutrophils were centrifuged at 100× *g* for 8 min then resuspended in RPMI supplemented with 1 mM CaCl_2_, 1 mM MgCl_2_, and 2% HI FBS at 1.67 × 10^6^ cells/mL and pretreated with indicated concentrations of MARCKS inhibitor peptides or PBS for 30 min at 37 °C with 5% CO_2_. A total of 450,000 neutrophils was aliquoted (300 µL) into BD Falcon polypropylene round-bottom tubes and allowed to settle for 10 min at room temperature. Immediately prior to the experiment, STm expressing GFP (GFP-STm) were opsonized with RPMI + 10% pooled bovine serum for 30 min at 37 °C then used to induce phagocytosis by stimulating neutrophils with STm (MOI 25:1) for the indicated amount of time. As a negative control, neutrophils were incubated with STm on ice for the duration of time [41,42]. At the end of incubation, 40% PFA was added to the tube to reach a final concentration of 2%, and the tube was placed on ice for 10 min prior to dilution with 1 volume of PBS [39]. Tubes were stored at 4 °C until ready to prep for acquisition. Tubes were centrifuged at 1100× *g* for 10 min, supernatants gently dumped, resuspended in 200 µL 0.125% Trypan blue in PBS, and transferred to a polystyrene tube then immediately acquired on a BD LSRII cytometer [43]. The neutrophil population was gated by a forward-scatter versus side-scatter plot to discriminate from cell debris. Singlet cells were included by gating based on side-scatter area versus side-scatter width and forward-scatter area versus forward-scatter width. Neutrophils not exposed to STm were used as a negative control for GFP-negative cells. Data from at least 7000 events gated on singlet neutrophils were collected. FCS files were analyzed by FlowJo version 10.8.1 (Ashland, OR, USA). The percentage of GFP-positive neutrophils was calculated using histograms. The fluorescence intensity was divided into two peaks: the GFP-negative cells that did not have phagocytosed bacteria (left peak) and phagocytosing GFP-positive cells (right peak) [44]. Flow cytometry experiments were performed in the Flow Cytometry and Cell Sorting facility at North Carolina State University—College of Veterinary Medicine. 

### 2.10. Phagocytosis—Microscopy

The neutrophil and GFP-STm co-culture was prepared as described for flow cytometry and incubated for 75 min. Neutrophils were transferred to Eppendorf tubes, centrifuged at 160× *g* for 5 min, supernatant discarded, and pellet resuspended in 200 µL PBS. Neutrophils were fixed for 10 min on ice with PFA (2% final concentration), then stained with 2 µg/mL AF954-conjugated wheat germ agglutinin (WGA) (Thermo Fisher) at room temperature for 10 min. Neutrophil suspensions were diluted with an additional 200 µL PBS, and 200 µL were centrifuged onto glass slides using a cytospin. Slides were air dried, and cover glass was mounted with DAPI Flouromount (Southern Biotech, Fisher Scientific, Hampton, NH, USA) before imaging with an Olympus IX83 inverted microscope (Olympus Life Science, Waltham, MA, USA). 

### 2.11. Respiratory Burst

Clear-bottom black 96-well plates were coated overnight with 5% HI FBS in HBSS and washed once with PBS prior to the experiment. Isolated primary neutrophils were resuspended to 1.0 × 10^7^ cells/mL in HBSS and primed with 10 ng/mL GM-CSF for 30 min at room temperature protected from light. Neutrophils were centrifuged at 100× *g* for 8 min and resuspended in RPMI supplemented with 10% autologous serum, 1 mM CaCl_2_, and 1 mM MgCl_2_. Neutrophils (150,000) were added to Eppendorf tubes containing indicated concentrations of MANS peptide, RNS peptide, or 10 µM DPI and incubated for 30 min at 37 °C with 5% CO_2_. Neutrophils were added to the plate, allowed to settle for 10 min, and 10 µM DHR-123 was added to each well. PBS or STm (MOI 50:1) was added to the plate and fluorescence was immediately read every 15 min for 2 h at 485 nm excitation, 525 nm emission. A control consisting of DHR-123 added to media alone was used for normalization prior to determining the fold change from unstimulated control. 

### 2.12. Salmonella Survival

Bacteria were grown as indicated above. Bacteria were pelleted and resuspended in HBSS with 1 mM CaCl_2_, 1 mM MgCl_2_, and 50% Bovine Serum and incubated at 4 °C shaking at 100 rpm for 30 min. Cultures were then pelleted and resuspended in HBSS with 1 mM CaCl_2_ and 1 mM MgCl_2_ at 5 × 10^7^ CFU/mL based on OD_600_. Bovine neutrophils and RPMI media controls were distributed in 100 µL volume on Nunclon-treated 96-well flat-bottomed plates. The plate was centrifuged at 100× *g* for 8 min at 4 °C. Peptides or PBS control were added to neutrophil and control wells for a final 10 µM or 100 µM peptide concentration and incubated for 30 min at 37 °C with 5% CO_2_. Following neutrophil incubation with peptides or PBS control, STm was added to wells at an MOI of 1:1, and the plate was incubated for 1 h at 37 °C with 5% CO_2_. Samples were then harvested with serial dilution and plating to enumerate CFU.

### 2.13. Statistical Analyses

All results were analyzed using GraphPad Prism software version 9 (San Diego, CA, USA) using repeated measures one-way ANOVA unless otherwise noted. Data are reported as mean ± SD with sample size indicated on each figure. The sample size indicates the number of individual donors tested in each assay. For all 96 well-based assays, two (*Salmonella* survival assays) or three technical repeats were averaged for the reported result. *p* values < 0.05 were considered statistically significant.

## 3. Results

### 3.1. Effect of MARCKS Inhibitor Peptides on Primary Neutrophil Viability

In this study, we planned to investigate the effects of two different peptide inhibitors of MARCKS: MANS and ED peptides. MANS peptide is a function-blocking peptide identical to the first 24 amino acids of MARCKS. Reported mechanisms of action for the MANS peptide include displacing MARCKS from the plasma membrane to the cytosol, altering cytoskeletal dynamics, and decreasing MARCKS phosphorylation [31,45,46]. ED peptide is a 37 amino acid peptide that mimics the effector domain of MARCKS. Reported mechanisms of action for ED peptide include decreasing MARCKS phosphorylation, direct binding to lipopolysaccharide (LPS), and acting as a nuclear localization signal and regulator of nuclear membrane PIP2. It also has downstream effects on gene expression [47,48,49]. MANS and ED peptides each have a corresponding control peptide, RNS and CP, respectively. The viability of peptide-treated neutrophils was evaluated using propidium iodide staining. Isolated primary neutrophils were treated with MANS, RNS, ED, and CP peptides. Neutrophil viability at 90 min was not significantly reduced by any of the tested concentrations of MANS peptide (Figure 1). Neutrophil viability at 90 min was significantly decreased by all tested concentrations of ED peptide (Figure 1). Further, ED peptide (without the presence of any other stimulus) induced the neutrophil respiratory burst (Appendix A). Given these results, the subsequent investigations of MARCKS inhibition on STm-induced neutrophil responses were focused on MANS-mediated MARCKS inhibition. 

### 3.2. MARCKS Inhibition Attenuates STm-Induced Neutrophil Adhesion

Neutrophil adhesion to the vasculature and to the basolateral membrane of the intestine is critical for neutrophil migration into the gut [50,51]. Previous work in our laboratory demonstrated an essential role for MARCKS in neutrophil adhesion [28,30]. Therefore, we hypothesized that MARCKS inhibition in primary bovine neutrophils would decrease STm-induced adhesion. Static neutrophil adhesion was induced using STm for 1 h. Pretreatment of neutrophils with MANS peptide for 30 min prior to stimulation with STm significantly attenuated adhesion in a concentration-dependent manner (Figure 2). The control peptide RNS had no effect on neutrophil adhesion. These data show an essential role for MARCKS protein function in STm-induced neutrophil static adhesion ex vivo.

### 3.3. MARCKS Inhibition Attenuates STm-Induced Neutrophil Migration 

Previous research demonstrated a clear role for MARCKS in neutrophil migration in vitro and in vivo [28,33]; therefore, we next evaluated the effect of MARCKS inhibition on STm-induced neutrophil migration. Neutrophil migration can be modeled using specialized ChemoTx plates where fluorescently labeled neutrophils are plated on top of a permeable membrane. Migration is then stimulated by the presence of chemoattractants (e.g., IL-8, fMLP, LTB_4_) or bacteria in wells below the membrane [24,28]. Migrated neutrophils are detected using fluorescence. Previous reports using this model for bacteria-induced migrations are limited [52,53]; therefore, two different approaches for stimulating STm-induced neutrophil migration were utilized. For the first approach, STm alone was used to stimulate neutrophil migration. For the second approach, knowing that IL-8 should enhance neutrophil migration [31], a low concentration of IL-8 (10 ng/mL) was added to the prepared STm.

STm alone induced significant neutrophil migration (~55%) compared to the unstimulated control (~25%). MANS peptide treatment significantly attenuated neutrophil migration in a concentration-dependent manner (Figure 3A). As expected, IL-8 significantly enhanced STm-induced neutrophil migration (~85% migration). Similarly, MANS peptide significantly attenuated migration induced by STm + IL-8 in a concentration-dependent manner (Figure 3B). RNS control peptide had no effect on neutrophil migration. These data show an essential role for MARCKS protein function in STm-induced neutrophil migration ex vivo. 

### 3.4. MARCKS Inhibition Enhances Neutrophil Phagocytosis of STm

Phagocytosis plays a key role in the immune response to bacterial infections. Previous studies have shown that neutrophils can effectively phagocytose both opsonized and non-opsonized Salmonella [54,55,56]. To determine the effect of MARCKS inhibition on STm phagocytosis, neutrophils were incubated with pooled serum-opsonized GFP expressing Salmonella (GFP-STm) for 15, 75, and 90 min. As a negative control, neutrophils were incubated with STm on ice [41,42]. The fluorescence of extracellular GFP-STm was quenched with Trypan blue before detection of phagocytosis using flow cytometry. The results demonstrate that MANS peptide treatment increased the number of GFP+ neutrophils at 75 and 90 min, indicating an increase in phagocytosis (Figure 4B,C). Furthermore, MANS peptide treatment also resulted in an increase in GFP+ MFI at 75 and 90 min (Figure 4E,F), suggesting that there was an increase in the number of ingested bacteria [57]. The cold control condition demonstrated lower phagocytic responses, as expected. The control peptide RNS had no effect on neutrophil phagocytosis of GFP-STm. The internalization of STm was confirmed using immunofluorescence microscopy (Figure 5). Therefore, the quantitative flow cytometry data and qualitative microscopy data demonstrate that MARCKS inhibition with MANS peptide enhances neutrophil phagocytosis of opsonized STm.

### 3.5. MARCKS Inhibition Enhances Neutrophil Respiratory Burst in Response to STm

Neutrophil reactive oxygen species (ROS) production, also termed respiratory burst, is a critical pathogen-directed host response. Previous research in our laboratory demonstrated that MARCKS inhibition with MANS peptide significantly inhibited immune complex-mediated ROS production [28]; therefore, we initially hypothesized that MARCKS inhibition would attenuate STm-induced neutrophil respiratory burst. However, the finding of increased phagocytosis with MANS treatment also suggested the possibility of enhanced ROS production. Intracellular STm-induced neutrophil respiratory burst was measured using DHR-123 for up to 2 h. Peak respiratory burst occurred around 90 min. The effect of MANS peptide treatment on respiratory burst was evaluated at specific time points and as total production (Figure 6A–D). MANS peptide treatment had no significant effect on STm-induced respiratory burst at 60, 75, and 90 min post stimulation. However, treatment with lower concentrations of MANS (25 µM and 10 µM) significantly enhanced total respiratory burst in response to STm, as measured by the area under the curve (Figure 6D). The control peptide RNS had no effect on respiratory burst. These data suggest that MARCKS inhibition with low concentrations of MANS peptide enhances STm-stimulated neutrophil respiratory burst ex vivo. 

### 3.6. MARCKS Inhibition Does Not Impact Survival of STm following Co-Incubation with Neutrophils 

Phagocytosis and respiratory burst are prominent tools for neutrophils to kill bacteria. However, STm is a facultative intracellular pathogen that can withstand reactive oxygen species. Therefore, we next investigated the effect of MANS peptide treatment on neutrophil-mediated STm killing. Neutrophils were pretreated with MANS or RNS peptide prior to co-incubation with STm for one hour. Treatment of neutrophils with either concentration of MANS (100 µM and 10 µM) or with RNS control peptide had no effect on STm survival (Figure 7). 

## 4. Discussion

MARCKS protein is known to play an important role in neutrophil effector functions. This investigation is the first to examine whether the inhibition of MARCKS function affects neutrophil responses to bacteria. Two MARCKS inhibitor peptides, MANS and ED, were used to inhibit MARCKS function in neutrophils. MANS is a function-blocking peptide that is known to displace MARCKS from the plasma membrane to the cytosol [31]. MANS peptide treatment has also been shown to decrease MARCKS phosphorylation, although the mechanism is unknown [45,58]. In addition to previous studies demonstrating MANS peptide inhibition of neutrophil migration, adhesion, and respiratory burst, the peptide also regulates LPS-induced inflammatory responses of macrophages [28,31,58]. The ED peptide and other peptides that mimic the MARCKS effector domain inhibit MARCKS phosphorylation across multiple cell types, presumably through competitive inhibition with protein kinase-C (PKC)-mediated MARCKS phosphorylation [47,59]. Prior studies with ED peptide or effector domain mimetic peptides have demonstrated success in suppressing cancer cell migration and metastasis in vitro and in vivo [60]. Other studies have highlighted the potential use of ED peptides in targeted delivery to cells for cell-specific killing mechanisms [61,62].

Adhesion is one of the first neutrophil functions in response to infection and inflammation. Therefore, we sought to determine whether MARCKS protein plays a significant role in STm-induced neutrophil adhesion. There are limited reports of whole bacteria-induced neutrophil adhesion [63,64]; however, STm induced significant static adhesion after 1 h of stimulation. The results demonstrate that MANS significantly attenuates neutrophil adhesion induced by STm, which is consistent with the results of neutrophil adhesion induced by other stimuli [28,31]. 

Previous reports have also identified MARCKS as a key player in neutrophil migration. *Salmonella*-induced migration was evaluated using two different experimental approaches. The first was STm alone and the second included STm + IL-8. This approach was taken due to limited evidence of bacteria alone stimulating migration in the ChemoTx plates. Further, IL-8 plays a prominent role in inducing neutrophil migration into the intestinal epithelium during *Salmonella* infection [65]. Both models induced significant and robust migration that was significantly attenuated with MANS peptide treatment. These findings are also consistent with previous reports identifying a role for MARCKS in ex vivo neutrophil chemotaxis [24,28].

Neutrophils are designed to kill bacteria through phagocytosis and subsequent release of ROS into the phagolysosome. Previous studies show that neutrophils phagocytose both opsonized and non-opsonized STm, and that infection with wild-type-LPS-expressing *Salmonella* results in the generation of reactive oxygen species (ROS) in TLR4-decorated, *Salmonella*-containing vacuoles [54]. The results show that inhibition of MARCKS with the MANS peptide enhances neutrophil phagocytosis of STm. Consistent with this result, we also found that treatment of primary bovine neutrophils with low concentrations of MANS peptide enhanced STm-stimulated neutrophil respiratory burst. This is somewhat surprising, given that previous research demonstrated that MARCKS inhibition (with the MANS peptide) either attenuated insoluble immune-complex-induced respiratory burst or had no effect on PMA-induced respiratory burst [28]. 

*Salmonella* benefits from neutrophil reactive oxygen species to enhance its colonization of the gastrointestinal tract, and the bacterium is known to induce respiratory burst in human neutrophils [39,66]. We sought to determine whether MARCKS protein played a role in STm-induced respiratory burst. Previous findings demonstrated that STm-induced neutrophil respiratory burst required whole serum [39]. We utilized autologous serum in the respiratory burst experiments to eliminate autofluorescence issues observed with purchased whole bovine serum and to provide complement opsonins for maximal respiratory burst. There was marked variability in respiratory burst in our study population, which could be influenced by the use of autologous serum, varying age, gestation stage, or prior exposure of the neutrophil donors to *Salmonella*. We also observed relatively low levels of respiratory burst activity compared with our previous reports in STm-stimulated human neutrophils [39]. 

The finding that MARCKS inhibition enhanced neutrophil respiratory burst and phagocytosis of STm is novel. This is the first report of enhancement of neutrophil effector functions with MANS-mediated MARCKS inhibition. In previous studies using MARCKS^−/−^ and wild-type fetal liver-derived murine macrophages, Carballo et al. found that MARCKS^−/−^ macrophages had reduced phagocytosis of zymosan compared to wild-type macrophages [67]. The differences between our findings and this previous report could be due to differences in stimulus, differences in the phagocytic index, and/or differences between a function-blocking peptide versus protein knockout. Enhanced neutrophil phagocytosis and ROS production with MARCKS inhibition could suggest that MARCKS has a negative regulatory role in these functions or cooperates with a different negative regulator protein. Known negative regulators of phagocytosis include the Src kinases Fgr and src homology 2 (SH2)-containing inositol phosphatase (SHIP) [68,69]. Further studies are needed to determine how MARCKS contributes to the regulation of *Salmonella*-induced neutrophil phagocytosis and respiratory burst. 

On a molecular level, it is possible to speculate how MARCKS inhibition may enhance neutrophil phagocytosis. During early phagocytosis, PIP2 is synthesized, accumulates at the phagocytic cup, and may control actin assembly at the phagosome. Phospholipase C (PLC) degrades PIP2, producing the lipid messenger diacylglycerol (DAG). Further cytoskeletal changes occur due to either a decrease in PIP2 or an increase in DAG. At the same time, phosphatidylinositol 3-kinase (PI3K) converts to PIP2 to phosphatidylinositol 3,4,5-trisphosphate (PIP3), triggering the closure of the phagosome [70], while at the plasma membrane, MARCKS is known to sequester 3–4 PIP2 molecules. Therefore, when MARCKS inhibition with MANS displaces MARCKS protein from the plasma membrane to the cytosol, there is an increase in PIP2 at the plasma membrane. Given that PIP2 plays such an important role in phagocytosis, it is possible that the increase in PIP2 availability is the reason for enhanced phagocytosis during MARCKS inhibition. It is also possible that peptide-mediated MARCKS inhibition contributes to cytoskeletal changes that make neutrophils more susceptible to invasion by *Salmonella* [71]. Further work is needed to fully elucidate how these two different processes of Salmonella internalization, *Salmonella* invasion vs. neutrophil phagocytosis, contribute to the increases in intracellular bacteria seen with inhibition of MARCKS protein.

Neutrophils utilize both β_2_-integrins (also known as complement receptor 3—CR3) and Fc receptors to mediate the engulfment of pathogens. In the current study, STm were serum-opsonized prior to stimulation, which likely initiated a phagocytosis mediated by both CR3 and Fc receptors [54,55]. However, we did not use any blocking antibodies to determine if either of the receptors had a dominant effect. In addition to Fc receptor and CR3 recognition of *Salmonella*, toll-like receptors 2, 4, and 5 are also important in the recognition and internalization of *Salmonella* [54,72,73,74]. TLR2/4 stimulation is also known to activate the β_2_-integrin Mac-1 [75]. Thus, it is apparent that neutrophil recognition of *Salmonella* results in the activation of β_2_-integrins either directly or indirectly through Fc receptors or toll-like receptor signaling. The finding that MARCKS inhibition attenuated both neutrophil adhesion and migration is consistent with our previously published results. Both adhesion and migration are known to be β_2_-integrin-dependent, and we have previously ascribed a role for MARCKS in neutrophil β_2_-integrin-dependent neutrophil functions [28] and outside-in activation [30,76]. LPS stimulation induces β_2_-integrin-dependent cell spreading [77], reinforcing the likelihood that *Salmonella* induces β_2_ integrin-dependent adhesion. Although our current investigation did not utilize specific β_2_-integrin inhibitors, we hypothesize that diverging effects of MARCKS inhibition on STm-induced neutrophil responses could be explained by the difference between β_2_-integrin-dependent (adhesion and migration) and independent (phagocytosis and respiratory burst) neutrophil effector functions induced by *Salmonella*. 

Formation of the Nicotinamide adenine dinucleotide phosphate-oxidase/Nox2 (NADPH) oxidase system and subsequent respiratory burst is intimately associated with neutrophil phagocytosis. Assembly of the five subunits of the phagocyte oxidase complex (gp91phox, p22phox, p40phox, p47phox, and p67phox) can be detected as early as 30 s after the onset of phagocytosis [78,79,80]. In light of this connection between these mechanisms, it seems plausible that the slight enhancement of ROS production that we observed is simply a predictable sequela to the enhanced phagocytosis caused by inhibition of MARCKS with the MANS peptide. However, the enhanced phagocytosis and respiratory burst had no impact on the survival of *Salmonella* co-incubated with neutrophils. Previous studies also demonstrate the survival of STm in the presence of neutrophils through various mechanisms [19,81]. Because phagocytosis is typically considered a mechanism for neutrophils to kill bacteria, it was surprising that MARCKS inhibition increased phagocytosis with no impact on STm survival. However, given that STm is tolerant of ROS, it is plausible that our results are a direct consequence of this tolerance [82]. Additional studies are needed to determine the mechanism by which MARCKS inhibition with MANS increases *Salmonella* survival in the presence of neutrophils.

Our results show that MARCKS protein function plays a role in multiple neutrophil effector responses to *Salmonella* Typhimurium ex vivo. Future investigations will determine whether this difference is due to differences in β_2_-integrin-dependent versus independent neutrophil responses to *Salmonella*. From a novel target standpoint, our results regarding MARCKS inhibition as a way to limit neutrophil-mediated host damage are promising. MARCKS inhibition may offer a strategy to limit excess neutrophil recruitment to sites of infection or inflammation through inhibition of adhesion and migration, while still preserving neutrophil host defense mechanisms such as phagocytosis and respiratory burst. 

In this study, we utilized a peptide inhibitor known as MANS to inhibit MARCKS function. While this short synthetic peptide worked well as a tool for our ex vivo experiments, it would likely need to be modified to avoid rapid degradation in vivo. Additionally, the novel findings reported here identify a role for MARCKS in neutrophil responses to *Salmonella*. Future studies are needed to further elucidate the role of MARCKS in neutrophil responses to other bacteria. 

## Figures and Tables

**Figure 1 biomedicines-12-00442-f001:**
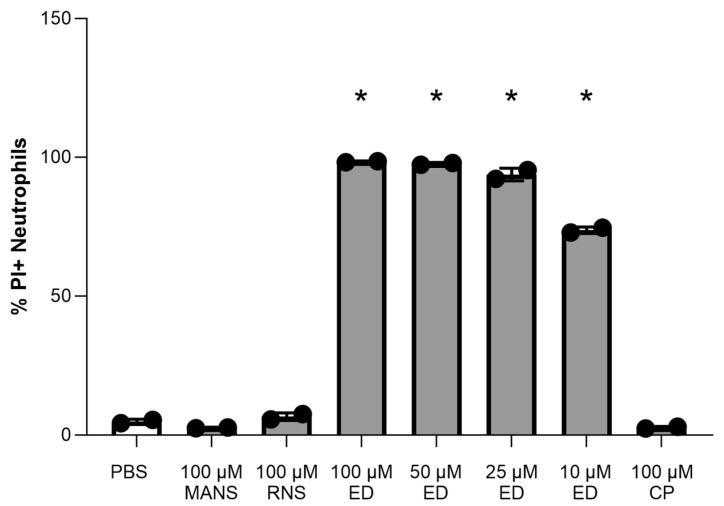
Treatment of primary neutrophils with ED peptide compromises cell viability. Neutrophils were treated with indicated concentrations of peptide or PBS (untreated) for 90 min, stained by propidium iodide, and then analyzed by flow cytometry. Each dot represents a single individual (n = 2), and data are represented as mean ± SD. Significant difference from PBS by repeated measures one-way ANOVA with Dunnett’s multiple comparisons is indicated by * (*p* < 0.05).

**Figure 2 biomedicines-12-00442-f002:**
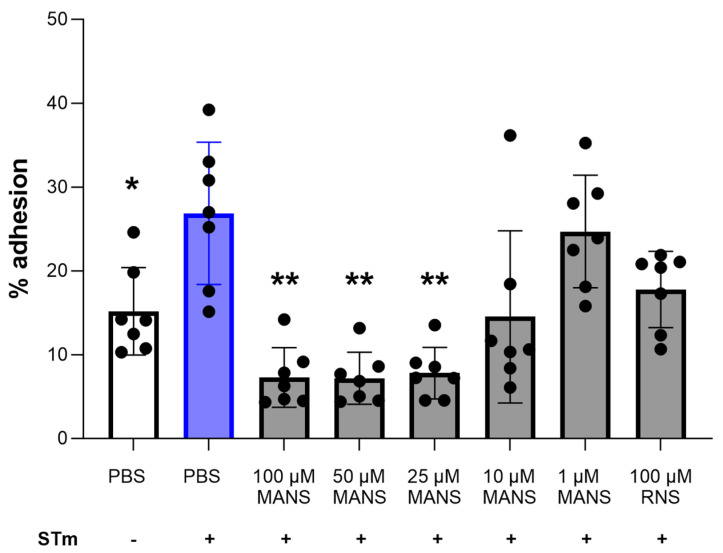
Treatment of primary bovine neutrophils with MANS peptide attenuates STm-induced neutrophil adhesion. Neutrophils were pretreated with indicated concentrations of MANS or RNS peptide and then labeled with calcein. Calcein-loaded neutrophils were stimulated with STm at MOI 50:1 for 1 h prior to serial washing and fluorescence measurements. Each dot represents a single individual (n = 7), and data are represented as mean ± SD. Significant difference from PBS + STm group by repeated measures one-way ANOVA with Dunnett’s multiple comparisons is indicated by * (*p* < 0.05) and ** (*p* < 0.01).

**Figure 3 biomedicines-12-00442-f003:**
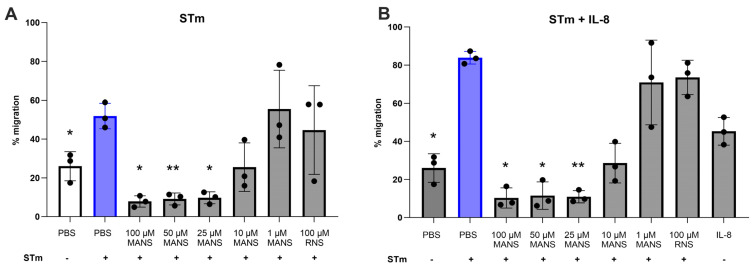
Treatment of primary bovine neutrophils with MANS attenuates STm- and STm+IL8-induced neutrophil migration. Calcein-loaded neutrophils were pretreated with indicated concentrations of MANS or RNS peptide and placed on top of the membrane in a ChemoTx plate and either STm (MOI 50:1) (**A**) or STm (50:1) + 10 ng/mL IL-8 (**B**) were added to the bottom wells to induce migration. The plate was incubated for 1 h at 37 °C and 5% CO_2_. The neutrophils on top of the membrane were removed, and the fluorescence of migrated neutrophils was determined. Each dot represents a single individual (n = 3), and data are represented as mean ± SD. Significant difference from PBS + STm group by repeated measures one-way ANOVA with Dunnett’s multiple comparisons is indicated by * (*p* < 0.05) and ** (*p* < 0.01).

**Figure 4 biomedicines-12-00442-f004:**
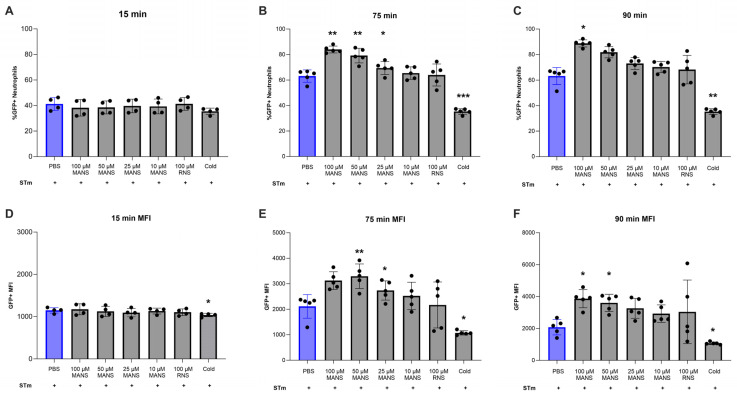
Treatment of primary bovine neutrophils with MANS peptide enhances neutrophil phagocytosis of GFP-STm. MANS or RNS-treated neutrophils were incubated with opsonized GFP-STm (25:1) for indicated times at 37 °C with 5% CO_2_. A cold control containing neutrophils and GFP-STm was included as a negative control for phagocytosis. The percentage of GFP-positive neutrophils was determined (**A**–**C**), and mean fluorescence intensity (MFI) was determined using FlowJo analysis (**D**–**F**). Each dot represents a single individual (n = 5), and data are represented as mean ± SD. Significant difference from PBS + STm group by repeated measures one-way ANOVA with Dunnett’s multiple comparisons is indicated by * (*p* < 0.05), ** (*p* < 0.01), *** (*p* < 0.001).

**Figure 5 biomedicines-12-00442-f005:**
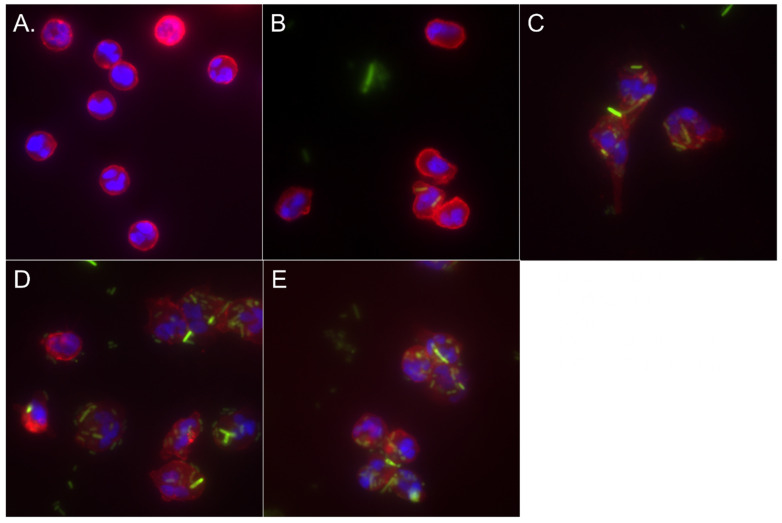
Neutrophil phagocytosis of GFP-STm. MANS or RNS peptide-treated neutrophils were incubated with opsonized GFP-STm (25:1) (green) for 75 min at 37 °C with 5% CO_2_, and then the plasma membrane was stained with wheat germ agglutinin (WGA) (red) and nuclei with DAPI (blue). (**A**) unstimulated neutrophils, (**B**) neutrophils + GFP-STm kept on ice (cold control), (**C**) STm + PBS (no peptide), (**D**) neutrophils treated with 100 µM MANS before STm stimulation, and (**E**) neutrophils treated with 100 µM RNS before STm stimulation. Representative image of three independent experiments.

**Figure 6 biomedicines-12-00442-f006:**
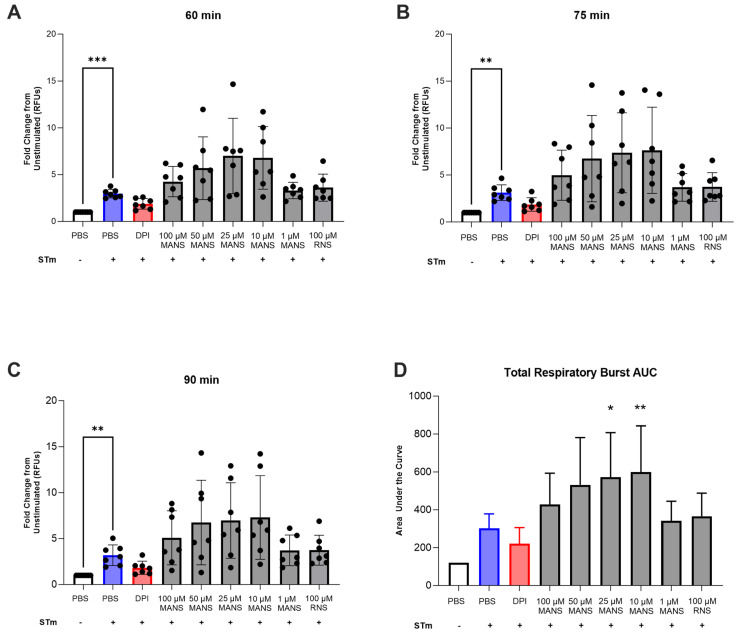
Treatment of primary bovine neutrophils with low concentrations of MANS peptide enhances STm-induced neutrophil respiratory burst. Neutrophils were pretreated with indicated concentrations of MANS or RNS peptide or, as a positive control of inhibition, neutrophils were pretreated with NADPH oxidase inhibitor DPI. DHR-123 was added to wells, and neutrophils were stimulated with STm (MOI 50:1). Fluorescence was measured for 2 h, and fold change from unstimulated control was calculated for 60 min (**A**), 75 min (**B**), and 90 min (**C**) of stimulation. The area under the curve was calculated (**D**). Each dot represents a single individual (n = 7), and data are represented as mean ± SD. Significant difference from PBS + STm group by repeated measures one-way ANOVA with Dunnett’s multiple comparisons is indicated by * (*p* < 0.05), ** (*p* < 0.01), and *** (*p* < 0.001).

**Figure 7 biomedicines-12-00442-f007:**
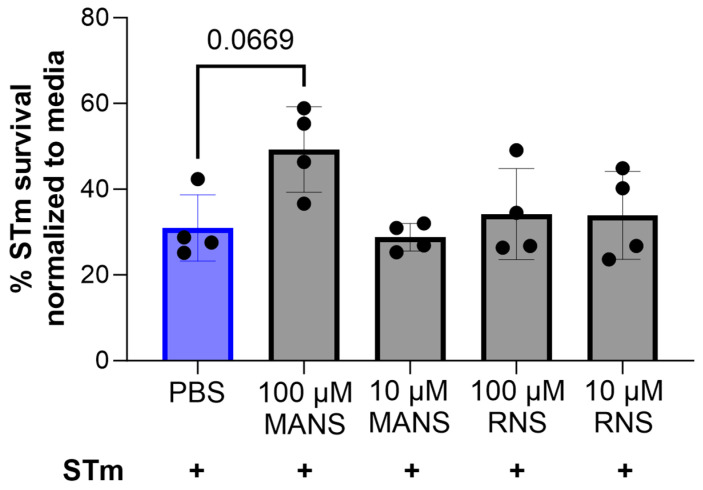
MARCKS inhibition with MANS peptide has no impact on neutrophil killing of STm. Each dot represents a single individual (n = 4), and data are represented as mean ± SD. Difference from PBS + STm group was calculated by repeated measures one-way ANOVA with Dunnett’s multiple comparison.

## Data Availability

The raw data supporting the conclusions of this article will be made available by the corresponding author upon request.

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
