# Peer review of "MARCKS Inhibition Alters Bovine Neutrophil Responses to Salmonella Typhimurium"

_biomedicines, 2024, doi:10.3390/biomedicines12020442_

Round 1
Reviewer 1 Report
Comments and Suggestions for Authors
The study by Conley et al. is very complex, well-written and conducted, providing solid proof of concept data on Myristoylated Alanine Rich C-kinase Substrate as a potential therapeutic target for Salmonella enterocolitis. Overall, the introduction sections are well-written and nearly cover the context and importance of the study. The "Materials and Methods" section is comprehensive, ensuring the study's reproducibility and offering a clear understanding of the techniques used. The presented results add credibility to the study. The discussion section effectively interprets the results and provides a deeper understanding of their significance.
Therefore, I found the manuscript data important in its field, and I support its further processing after appropriate minor modifications as outlined below:
L2: MARCKS – please avoid the direct use of acronyms in the title. Revise!
L17, L20: “We hypothesized” “our data”, etc. – please avoid the personal mode verb formulations, may sound unprofessional. Revise this concern throughout the manuscript!
L29: “…oxygen species (ROS).” – In the introduction section, please ensure the using of appropriate citations after each statement. Furthermore, please mention and enhance the documentation about the importance of Salmonella enterica for human medicine.
L64-66: according to the MDPI journal requirement, when you present reagents or devices please uniformly mention their production company name, city, and country throughout the materials and methods section.
L89-90: “All cows were considered healthy.” – This is a very general statement, considering the study aim. Can you provide some details on how was this issue evaluated? The same concern for line 110 “10 healthy”
L540: please provide a separate Conclusion section and mention the study limitations.
Reviewer 2 Report
Comments and Suggestions for Authors
The data are intriguing but provide no mechanistic insights.
The authors show that a myristoylated-MARKS peptide attenuates Salmonella Typhimurium (STm)-mediated adhesion and migration while increasing phagocytosis and production of reactive oxygen species (ROS).
General comment: It is a well-written manuscript. The data are intriguing but provide no mechanistic insights. The fact that MAMS peptide reduces cell adhesion could introduce a bias. Can the authors repeat the phagocytosis assay under synchronized conditions (centrifugation at 170 g for 5 min)?
Specific criticisms:
Why not use IL-8 as a control to determine whether MAMS peptide reduces neutrophil migration? Of course, IL-8 will induce neutrophil migration. However, mixing IL-8 and Salmonella Typhimurium (STm) adds very little insight into the mechanism through which MAMS inhibits adhesion.
Figure 4 should include the phagocytosis data at 15 min. The observation that MANS can promote STm phagocytosis is intriguing. Is the effect of MANS restricted to serum-opsonized STm or enhanced phagocytosis observed with other opsonized particles (IgG-opsonized E. Coli, serum-opsonized zymosan)?
Figure 5. The authors must include a quantitative assessment of the STm-positive phagosome to support their conclusion.
Is STm-induced ROS production signification inhibited by DPI?
The supplemental figure 1 requires a figure legend.
Round 2
Reviewer 2 Report
Comments and Suggestions for Authors
I have no further comments.